# Experience of Kenyan researchers and policy-makers with knowledge translation during COVID-19: a qualitative interview study

Fatuma Hassan Guleid [ORCID],[1] Alex Njeru,[1] Joy Kiptim,[1] Dorcas Mwikali Kamuya,[2,3] Emelda Okiro,[3,4] Benjamin Tsofa [ORCID],[5] Mike English [ORCID],[3,6] Sassy Molyneux,[2,3] David Kariuki,[7] Edwine Barasa[3,8]

For numbered affiliations see end of article.

**Correspondence to**
Ms Fatuma Hassan Guleid;
fguleid@kemri-wellcome.org

## ABSTRACT

**Objectives** Researchers at the KEMRI-Wellcome Trust Research Programme (KWTRP) carried out knowledge translation (KT) activities to support policy-makers as the Kenyan Government responded to the COVID-19 pandemic. We assessed the usefulness of these activities to identify the facilitators and barriers to KT and suggest actions that facilitate KT in similar settings.

**Design** The study adopted a qualitative interview study design.

**Setting and participants** Researchers at KWTRP in Kenya who were involved in KT activities during the COVID-19 pandemic (n=6) were selected to participate in key informant interviews to describe their experience. In addition, the policy-makers with whom these researchers engaged were invited to participate (n=11). Data were collected from March 2021 to August 2021.

**Analysis** A thematic analysis approach was adopted using a predetermined framework to develop a coding structure consisting of the core thematic areas. Any other theme that emerged in the coding process was included.

**Results** Both groups reported that the KT activities increased evidence availability and accessibility, enhanced policy-makers' motivation to use evidence, improved capacity to use research evidence and strengthened relationships. Policy-makers shared that a key facilitator of this was the knowledge products shared and the regular interaction with researchers. Both groups mentioned that a key barrier was the timeliness of generating evidence, which was exacerbated by the pandemic. They felt it was important to institutionalise KT to improve readiness to respond to public health emergencies.

**Conclusion** This study provides a real-world example of the use of KT during a public health crisis. It further highlights the need to institutionalise KT in research and policy institutions in African countries to respond readily to public health emergencies.

## INTRODUCTION

The COVID-19 pandemic has required policy-makers to make decisions that have significant health, social and economic consequences. Incorporating the best available research evidence to guide decision-making

## STRENGTHS AND LIMITATIONS OF THIS STUDY

⇒ This study employed in-depth interviews to elucidate the engagement between researchers and policy-makers and how evidence is used in a pandemic.
⇒ The study provides lessons that can inform knowledge translation practice in similar settings.
⇒ Including researchers as participants in the study may have introduced bias.

is essential for an effective response. However, public health emergencies (PHEs) present several challenges to evidence-informed decision-making (EIDM). These include: (1) a rapidly increasing and evolving evidence base; (2) increased uncertainty and risk; and (3) sheer time pressure to make quick decisions.[1] Knowledge translation (KT) is best placed to respond to this challenge.[2 3] The WHO defines KT as 'the synthesis, exchange and application of knowledge by relevant stakeholders to accelerate the benefits of global and local innovation in strengthening health systems and improving people's health'. There are four key models of KT[4]: (1) push models where researchers disseminate research evidence to those who can use it; (2) pull models where research users ask for evidence; (3) exchange models which depend on partnerships between researchers and research users for mutual benefit; and (4) integrated models where linkages and exchange occurs across institutions and systems.

There are several documented barriers to effective KT in low-middle-income regions, such as Africa. For researchers, these include lack of capacity to plan and implement KT, poor relationships between researchers and policy-makers, competing demands for time,

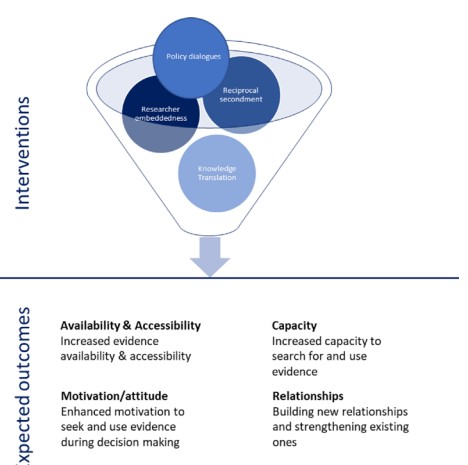

**Figure 1** Framework for assessing the usefulness of enhancing evidence-informed decision-making for health activities.

a lack of institutional incentives to participate in KT, the complexity of the policy process and difficulty in communicating research evidence.[5–7] For policy-makers, the most frequent barriers include lack of accessible evidence, lack of timely research output and poor capacity to appraise and understand research evidence.[8] These barriers are exacerbated in the context of PHEs. While there have been several studies on KT in Africa, the literature on the use of KT in responding to PHEs in Africa is limited. More research is needed to inform how institutions in Africa can conduct KT effectively in emergency settings.

### Enhancing evidence-informed decision-making for health (EEVIDEM)

The EEVIDEM project was a KEMRI-Wellcome Trust Research Programme (KWTRP) initiative based on the KT push, exchange and integrated models that sought to promote EIDM in Kenya using four interventions. The first intervention was researcher embeddedness into policy-making spaces, such as technical working groups. The second intervention was reciprocal secondment. In this arrangement, a policy-maker from the Ministry of Health (MoH) spends 50% of their working hours at KWTRP, allowing them to participate in research activities. Likewise, one researcher from KWTRP spends 50% of their time at the MoH. The third intervention was hosting regular policy dialogues. The fourth intervention was knowledge synthesis and dissemination (hereafter collectively referred to as 'KT activities'). This intervention involved generating evidence and disseminating KT products to policy-makers and is the paper's focus.

While KWTRP had engaged in KT activities before EEVIDEM, these were ad hoc, inconsistent and driven by few individual researcher initiatives. Institutionalising KT means taking steps to ensure that the systems, structures, resources and support needed for KT to become standard practices are available. The EEVIDEM project supported the institutionalisation of KT within KWTRP by facilitating the development of a policy engagement

and KT strategy, creating mechanisms for documentation and evaluation of KT efforts, providing financial resources for KT, staffing with KT capacities, formation of structures that support linkages between researchers and policy-makers, and providing technical support to researchers to develop and implement KT strategies.

We sought to describe the experiences of the researchers and policy-makers involved in EEVIDEM's KT activities that supported the COVID-19 response and assess the influence of these activities. The lessons learnt from these experiences can strengthen the processes that promote evidence-informed responses during PHEs in African settings.

## METHODS
### Study setting

In Kenya, the MoH has established organisational structures to support using evidence in policy-making. The Kenya Medical Research Institute (KEMRI), of which KWTRP is a part, is mandated to generate this evidence. During the pandemic, the MoH established the National COVID-19 taskforce. Its membership was drawn from the MoH and other government agencies, research and academic institutions, United Nations agencies, civil society organisations and the private sector. The taskforce coordinates and mobilises technical advice to the MoH and advises other ministries on appropriate measures. The taskforce's action points are implemented within the MoH's departmental structure, which includes the Director General's Office and the Emergency Operations Centre. KWTRP researchers provided research evidence and engaged with policy-makers within these structures.

### EEVIDEM conceptual framework

To guide the assessment of the influence of KWTRP's KT activities on evidence uptake during the COVID-19 pandemic, we adapted a determinant framework (figure 1) that conceptualises determinants that have been found to influence KT outcomes.[8–10] This framework emphasises four overlapping and interacting determinants, namely: (1) availability and accessibility of evidence; (2) attitude/motivation to search for and use evidence; (3) capacity to appraise and use evidence; and (4) the relationship between researchers and policy-makers.

### KT activities

Initially, the researchers synthesised evidence on questions they thought were of high priority. As the pandemic progressed, policy-makers started engaging with researchers to identify priority questions and coproduce COVID-19 evidence. The evidence was packaged as rapid response evidence briefs and/evidence summaries. Dissemination was primarily through email, and in some cases, researchers were invited to do oral presentations.

**Table 1** Institutions and departments of participants in the study

| Researchers | Policy-makers |
|---|---|
| Health Economics Research Unit | Ministry of Health |
| Health Systems and Policy Research Group | Ministry of Health—Emergency Operations Center (EOC) |
| Health Services Research Unit | Ministry of Health |
| Virus Epidemiology and Control Research Group | Presidential Policy and Strategy Unit |
| Virus Epidemiology and Control Research Group | Ministry of Health—EOC |
| Policy Engagement and Knowledge Translation | World Health Organisation—Kenya Office |
| | Ministry of Health—EOC |
| | Ministry of Health—Office of the Cabinet Administrative Secretary of Health |
| | Ministry of Health—Office of the Director-General of Health |
| | Ministry of Health—Department of Health Policy & Research |
| | Kilifi County Department of Health Services |

### Study design

This study adopted a qualitative interview cross-sectional design. A total of 17 key informant interviews (KIIs) were carried out on two groups: researchers and policy-makers (table 1). All KWTRP researchers involved in KT activities (n=6) to support the COVID-19 response were interviewed until data saturation was achieved. The selected researchers identified the policy-makers they engaged during their KT activities who were then invited to participate in the study. We identified 21 eligible policy-makers. Out of this, 11 policy-makers were interviewed, after which a point of information saturation was reached. EEVIDEM's conceptual framework was used to develop the interview guides and facilitate the interviews. Separate guides were used for researchers and policy-makers. Two researchers (FHG and AN) conducted all the interviews virtually using Microsoft Teams after each participant provided informed consent. Each interview lasted between 30 and 60 min and was audio-recorded using an encrypted audio recorder, except for one policy-maker who declined consent to audio recording—the interview was captured by a note-taker in handwritten notes. Data were collected from March to August 2021.

### Patients and public involvement

There was no patient or public involvement in the study.

### Data management and analysis

Audio recordings were transcribed verbatim in MS Word. The transcripts were coded using NVIVO qualitative data

**Table 2** Summary of briefs

| Topic | Number |
|---|---|
| Seroprevalence and genomic surveillance of SARS-CoV-2 | 21 |
| Other COVID-19-related briefs | 26 |
| Clinical surveillance | 9 |

analysis software (QSR International, V.12, 2014). For analysis, a predetermined coding structure consisting of the core thematic areas of the conceptual framework was used. Any other theme that emerged in the coding process was included. Code reports were generated and analysed once all data were coded in their respective themes (FHG and AN).

## RESULTS

Fifty-six rapid response evidence briefs and summaries were developed. These briefs shared evidence from primary studies, including modelling studies, genomic surveillance of SARS-CoV-2, seroprevalence of SARS-CoV-2 antibodies, health systems assessment and economics studies, clinical surveillance, and the synthesis of existing evidence on topics requested by policy-makers (table 2). Briefs were 2–4 pages long and included key messages, brief background and evidence findings.

Participants described several facilitators and barriers that shaped their experience, as summarised in table 3.

### Accessibility and availability

The researchers felt that their KT efforts increased evidence availability and accessibility. A significant facilitator of this was the support of existing KT capacity in KWTRP through EEVIDEM that provided guidance and supported researchers in the consistent and timely development of KT products.

> It has been the existence of the structures that we've managed to create with this platform [EEVIDEM]. It's something that we don't emphasise enough, having that pool of people. KII-R05, Researcher

Policy-makers also reported that the briefs enhanced their access to research evidence. They highlighted the format of the briefs as important in making evidence more accessible as they were concise and used simple language.

> I found the format fairly useful. I like that there was a section summarising what the rest of the brief held. It was beneficial as one can quickly garner some of the key evidence from that particular brief very quickly. KII-PM08, Policy-maker

Both groups reported the importance of augmenting written briefs with an oral presentation of evidence, which provided them with an opportunity to discuss the evidence and reduce misinterpretation:

**Table 3** Facilitators and barriers of evidence uptake

| | Researchers | | Policy-makers | |
|---|---|---|---|---|
| | **Facilitators** | **Barriers** | **Facilitators** | **Barriers** |
| Availability and accessibility | ▶ Institutional support and existing KT capacity<br>▶ Oral presentation of evidence | ▶ Timeliness<br>▶ Limited availability of local data<br>▶ Lack of feedback mechanism | ▶ Overall easy to read briefs<br>▶ Oral presentations<br>▶ Easy access to researchers | ▶ Timeliness<br>▶ Unequal access to briefs<br>▶ Sometimes briefs were too technical or long |
| Motivation | ▶ Producing relevant evidence to policy-makers<br>▶ Availability of researchers to produce evidence | ▶ Risk of sharing evidence that clashed with policy-makers' expectations or views | ▶ When evidence answered a priority question<br>▶ Availability of researchers to share evidence<br>▶ The positive feedback loop that was generated when evidence was used to inform prior decisions<br>▶ Credibility of researchers | ▶ Sometimes researchers can be unresponsive/unavailable |
| Capacity | ▶ Consistent interaction with policy-makers and briefs increased capacity to understand and use evidence | ▶ Policy-makers lack time to take part in capacity development activities | ▶ Regular interaction with research evidence and researchers | ▶ Short staffing and increased responsibilities meant they did not have time for capacity building |
| Relationships | ▶ Regular sharing of credible, timely and relevant evidence | ▶ Constant personnel changes at the MoH<br>▶ Being viewed as political players<br>▶ Building and maintaining relationships was time-consuming | ▶ Leveraging existing informal relationships with researchers | ▶ When researchers failed to meet all of the policy-makers' expectations for collaboration |

KT, knowledge translation; MoH, Ministry of Health.

They [the researchers] present most of the information and not only presented in writing, they normally present there [at the MoH]. So you can always seek clarification should there be a need to. KII-PM01, Policy-maker

The presentations are best because they involve you interacting with the people who are making the decisions or who will interpret the evidence. So, you get to communicate far more information unlike these other modes of communication. KII-R03, Researcher

The main barrier to evidence availability and accessibility for both groups was timeliness. The time it took to generate and synthesise high-quality evidence was often longer than the time within which evidence was needed for timely policy-making.

Sometimes they are required urgently, and I have to turn them around while ensuring that they're of

reasonable quality which can be a challenge KII-R01, Researcher

The challenge that is experienced is the length of time from once a request is sent out for a certain matter to the time a brief is submitted on the same KII-PM01, Policy-maker

Other barriers mentioned by researchers included the limited availability of local data, making it difficult for researchers to adapt evidence to local settings. In addition, the lack of a feedback mechanism left some researchers unaware of how useful briefs were to policy-makers as they rarely got direct feedback. Policy-makers mentioned two more barriers. One was the length of the briefs; they felt that when a brief was long or too technical, it lowered the accessibility of the evidence. In addition, policy-makers shared that there was unequal access to evidence among them:

In my position, getting evidence is not as difficult; however, other officers within the directorate find it a bit difficult to access the evidence and are told write officially or show that it comes through like Director General's office so that they can access the evidence. KII-PM01, Policy-maker

Researchers highlighted three factors that could improve evidence availability and accessibility. These included expanding KT capacity in KWTRP, increasing availability of local data to researchers to support the contextualisation of evidence to the local setting, and more feedback from policy-makers to help researchers determine if policy-makers found evidence easier to use through their KT activities. Suggestions by policy-makers included making briefs even easier to read by using more illustrations and tables and depositing briefs on open platforms so that more policy-makers have access to evidence.

### Motivation/attitude
Both groups reported that regular sharing of briefs enhanced policy-maker motivation to seek and use research evidence. A key facilitator was the relevance of the evidence. Motivation was enhanced as long as the evidence answered a priority question:

> To a certain extent, the larger the crisis, the more the need for research. However, the endless loop for motivation would be the need to answer a question at the forefront of policy-makers' minds. KII-PM03, Policy-maker

Both groups also felt that the availability and willingness of researchers to generate and share evidence with policy-makers was an important motivator.

> I think just knowing that we are available to them is a huge motivator. That someone else can search for evidence and summarise it and then present it to them is a huge motivator. KII-R06, Researcher

Policy-makers mentioned other facilitators including the positive feedback loop generated when evidence-informed policies were implemented and prior engagement between the two groups. These factors enhanced policy-makers' confidence in the researcher's credibility.

For researchers, the barriers to enhancing motivation included the risk of sharing evidence that clashed with policy-makers' expectations. This could create conflict between the two groups and limit the researcher's agency.

> Sometimes, the evidence we generate is at odds with policy-makers' position or expectations. That puts us in uncomfortable situations. KII-R02, Researcher

To enhance the motivation of policy-makers, researchers felt that sustaining engagement with policy-makers would be necessary. Policy-makers suggested that researchers should sensitise more policy-makers on the importance of considering evidence during policy-making.

### Capacity
Both groups felt that participation in KT activities enhanced the capacity of policy-makers to use evidence. They felt that regular interaction with researchers and evidence increased policy-makers' ability to interpret and use evidence during policy-making:

> Working with researchers has built confidence in using information for making decisions, yes. KII-PM07, Policy-maker

A challenge identified by researchers in increasing the capacity of policy-makers to appraise and use research evidence is the lack of time for policy-makers to participate in capacity-building activities:

> We know that policy-makers do not have time. KII-R02, Researcher

Policy-makers also mentioned that short staffing in the department meant that they did not have the staffing available to continue training while attending to their everyday tasks.

Overall, both groups felt that more KT training and workshops to improve capacity to appraise and use evidence were needed when policy-makers were available.

### Relationships
Both groups reiterated that their relationships were valuable and had strengthened during the pandemic. For policy-makers, an important facilitator for this was informal relationships with researchers. This meant that researchers were more accessible to them. Some noted that their relationship with some of the researchers was established even before joining the MoH, but further engagement deepened this relationship:

> They weren't new to me, but of course, the deepening of the relationships has come about now by us engaging further on this pandemic response that much I would say. KII-PM01, Policy-maker

According to the researchers, consistently sharing KT products that were timely and relevant made policy-makers aware of researchers' value, which cultivated a good working relationship. However, several barriers to relationship building were identified. From the researchers' perspective, these include: (1) the constant personnel changes at the MoH; (2) being perceived as political players rather than evidence generators; and (3) building and maintaining relationships took time and effort which was usually not accounted for as part of their research activities. From the policy-makers' perspective, one barrier was the researchers' failure to meet all collaboration expectations. The policy-maker expected more capacity building but conceded that due to COVID-19 restrictions, there were fewer opportunities. Suggestions for addressing relationship barriers included strengthening institutional relationships and ensuring that engagement and communication between the two groups were sustained. Finally, researchers noted that

they needed training on communicating with policy-makers and navigating policy spaces without being seen as political.

### Influence of KT activities on the COVID-19 response

Participating researchers felt that the KT activities they were involved in were useful in informing the COVID-19 response. This was reflected in the way policy-makers regularly referred to briefs produced by the researchers and used evidence produced. According to the policy-makers, the evidence disseminated by the researchers was used to inform COVID-19 policy-making as noted by several policy-makers:

> Much of the evidence shared was whether we should be able to strengthen some areas and also reinforce lockdowns. Some of the measures put in place have come from the reports that researchers provide. KII-PM07, Policy-maker

> The briefs from genome sequencing have enabled us to map out hot spot areas which we used to give an advisory about movement. We also used the recommendations that came off the sero-surveillance evidence to advise the people in Nairobi on how to conduct themselves. KII-PM09, Policy-maker.

### DISCUSSION

The KT activities described here promoted evidence use for COVID-19 response. Some key facilitators of evidence uptake were highlighted. First were the tools used to communicate evidence, including rapid response evidence briefs and oral presentations. Similar to our finding, previous studies also report that the evidence briefs and summaries are well received by policy-makers and increase the likelihood of using them.[11–13] Second,

strong relationships between researchers and policy-makers were perceived as critical for KT activities as they enhanced evidence accessibility and built trust, similar to other studies.[8 14 15] Furthermore, policy-makers in this study shared that informal relationships with researchers facilitated their professional relationships during the COVID-19 response. Likewise, Hyder *et al* reported that when policy-makers knew researchers socially, they were more open to considering their research.[16]

Lack of time has consistently been identified as a critical barrier to evidence uptake.[8 14] Often, the amount of time required by researchers to generate high-quality evidence does not match the policy-makers' time frame for policy-making. In addition, researchers and policy-makers have other responsibilities and find it challenging to dedicate time to KT activities. Finally, building relationships required continuous engagement, which was time-intensive. Participants also reflected on the lack of skills to participate in KT and the insufficient institutional infrastructure to carry out KT. According to both groups, the main suggestion to address these barriers is institutionalising KT and expanding KT capacity at the research institution and MoH.

It was noted that institutions in LMICs are less prepared to conduct KT that responds to the pandemic.[17] The experiences and outcomes we report highlight the urgent need to institutionalise KT within research and government institutions in Africa to respond to PHEs adequately. For example, through institutionalisation, funds for KT can be planned for as part of an institution's budget. Such investments would strengthen KT capacity in both research and policy institutions. Researchers with increased capacity and access to structured mechanisms for KT would be more motivated to take part in KT. Likewise, policy-makers with increased capacity and

**Table 4** Suggestions to improve evidence uptake during PHEs using KT

| Researchers should | Policy-makers should |
|---|---|
| ► Ensure KT products are relevant to policy-makers' needs | ► Institutionalise KT within policy-making spaces |
| ► Improve readability of briefs by including tables and figures | ► Make local evidence more available for researchers to contextualise evidence better |
| ► Increase KT capacity to respond to evidence demands promptly | ► Actively engage with researchers to report on the usefulness of KT products and suggest areas of improvement |
| ► Improve communication skills | ► Strengthen and formalise institutional relationships with research institutions |
| ► Sensitise more policy-makers on the utility of research evidence | |
| ► Conduct more training and workshops on KT for policy-makers | |
| ► Create open repositories for evidence | |
| ► Maintain communication and interaction with policy-makers | |
| ► Institutionalise KT in research institutions | |

KT, knowledge translation.

institutional support can engage more effectively with research-type evidence. Institutionalisation can also overcome the problems with timeliness in terms of low staffing. Institutions can hire more KT experts to provide on-demand evidence promptly. This would put less pressure on researchers who balance their daily research activities while responding to new evidence demands. Institutionalisation also promotes relationship building and partnerships, especially with the regular staff turnover at MoHs. Such formal partnerships build trust and credibility and facilitate mutual understanding, which is critical when responding to crises. In our case, we leveraged on the EEVIDEM project that was already underway when the pandemic began. EEVIDEM provided a system that promoted a consistent exchange of knowledge and provided the resources, structures and incentives to support KT.

Institutionalisation of KT is particularly needed in African countries, where the use of evidence during decision-making is still challenging.[17] Research institutions in Africa are significantly contributing to COVID-19 research.[18–20] However, to ensure that this research is used to inform local responses, evidence use and KT should be institutionalised.

## Limitations

The researchers in this study also participated in the KT activities described here, which may impact analysis. However, the participatory nature of our KT approach made it inherently collaborative, which inadvertently set the researcher up as a participant. To address trustworthiness, the researchers were transparent about their relationship to the topic of the investigation and maintained reflexivity throughout the process. In addition, the analysis was presented to the participants for sense checking to see if their views were correctly represented. The study was also presented to broader forums to check for objectivity. Finally, we interviewed 17 participants, which limited statistical generalisability. However, as is consistent with qualitative studies, the aim was not statistical generalisability, but rather analytical generalisability, typically achieved with relatively small sample sizes that facilitate in-depth exploration of participants' perspectives.[21] Therefore, the analytical themes elucidated by the study are relevant to contexts similar to the study context.

## CONCLUSIONS

The key learning from this study is the importance of institutionalising KT in research and government institutions to support PHEs. In this study, we sought to increase evidence uptake by enhancing four main determinants of uptake: accessibility, motivation, capacity and relationships. We found that by using this framework, we had relative success in informing policy-making during COVID-19. The novelty of the approach lies in the context within which it is applied. By accurately recording our experiences, interventions and outcomes we can derive lessons

to inform the establishment of institutionalised KT that responds to PHEs (table 4).

## Author affiliations

[1]Policy Engagement & Knowledge Translation Unit, KEMRI-Wellcome Trust Research Programme Nairobi, Nairobi, Kenya
[2]Health Systems and Research Ethics, KEMRI-Wellcome Trust Research Programme, Kilifi, Kenya
[3]Nuffield Department of Medicine, University of Oxford, Oxford, UK
[4]Population Health, KEMRI-Wellcome Trust Research Programme Nairobi, Nairobi, Kenya
[5]Health Policy and Systems Research, KEMRI-Wellcome Trust Research Programme, Kilifi, Kenya
[6]Health Services Unit, KEMRI-Wellcome Trust Research Programme Nairobi, Nairobi, Kenya
[7]Department of Health Policy and Research, Ministry of Health, Nairobi, Kenya
[8]Health Economics Research Unit, KEMRI-Wellcome Trust Research Programme Nairobi, Nairobi, Kenya

**Contributors** FHG, AN, DMK, EO, BT, ME, SM, DK and EB conceived the protocol. FHG and AN participated in the data collection and extraction. FHG and AN performed data analysis and FHG, JK, DMK, EO, BT, ME, SM, DK and EB participated in the article writing. FHG acts as the guarantor of this article

**Funding** This manuscript is published with the permission of the director of KEMRI and was funded by a Wellcome Trust award (215745); additional funds from a Wellcome Trust core grant awarded to KWTRP (092654) supported this work. The funders had no role in study design, data analysis, decision to publish, drafting, or submission of the manuscript. The views expressed in the papers are for the authors and not for the organisations they represent.

**Competing interests** None declared.

**Patient and public involvement** Patients and/or the public were not involved in the design, or conduct, or reporting, or dissemination plans of this research.

**Patient consent for publication** Not applicable.

**Ethics approval** This study involves human participants and was approved by The Scientific and Ethics Review Committee of the Kenya Medical Research Institute. The approval number is KEMRI/RES/7/3/1. Participants gave informed consent to participate in the study before taking part.

**Provenance and peer review** Not commissioned; externally peer reviewed.

**Data availability statement** All data relevant to the study are included in the article or uploaded as supplementary information.

**ORCID iDs**
Fatuma Hassan Guleid http://orcid.org/0000-0002-8103-8136
Benjamin Tsofa http://orcid.org/0000-0003-1000-1771
Mike English http://orcid.org/0000-0002-7427-0826

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
