## [Reviewer comments · BMJ Open]

ARTICLE DETAILS

TITLE (PROVISIONAL)	Experience of Kenyan researchers and policy-makers with knowledge translation during COVID-19: a qualitative interview study
AUTHORS	Guleid, Fatuma; Njeru, Alex; Kiptim, Joy; Kamuya, Dorcas; Okiro, Emelda; Tsofa, Benjamin; English, Mike; Molyneux, Sassy; Kariuki, David; Barasa, Edwine

VERSION 1 – REVIEW

REVIEWER	Tina Purnat World Health Organization, Department of Pandemic and Epidemic Preparedness and Prevention
REVIEW RETURNED	25-Jan-2022

GENERAL COMMENTS	The paper reports important KT considerations from an LMIC setting. One of the conclusions is a recommendation to institutionalize KT to address slow generation of evidence to inform policy during emergencies. KT platforms are often slow even in non-emergency settings and not equipped to generate evidence during emergency response mode. I suggest that the authors discuss the recommendation in more detail - is there anything that would need to change in an KT platform as practiced now or when institutionalized that would enable faster feeding into the emergency response?
--

REVIEWER	David Foord University of New Brunswick
REVIEW RETURNED	01-Feb-2022

GENERAL COMMENTS	Date: February 1, 2022 From: Reviewer To: BMJ Open Re: Knowledge translation during COVID-19: Experience of Kenyan researchers and policymakers. Manuscript ID: bmjopen-2021-059501 As requested, please see my comments on the manuscript. 1. Thank you for the opportunity to review and provide comments on the manuscript. In general, the research topic is interesting and timely. My substantive comments focus, in part, on the need to better identify the research gap and novel contribution from the study. The authors introduce a number of promising concepts in the abstract of the paper and return to them briefly in the discussion section, but not enough was done with these concepts. I suggest they explore them in more depth if the proceed with a revision to the paper. My detailed comments are provided below. 2. In the abstract I wanted to know what was meant by
--

	“institutionalise KT”, “fully embed KT”, “core functions of research” and “structures that support KT”. I appreciate that these terms may not be able to be defined in the abstract given word limits, but I wanted to see them defined and examined in the paper. 3. In the introductory section I expected to learn about the relevance of the research to theory and practice. The stated relevance of the study is to address issues with the rapid increase in volume of COVID-19 evidence and misinformation. I did not think a strong case was made for these issues. 4. I also expected to see a definition of the core construct of knowledge translation (KT). One is given (synthesis, dissemination, and application of evidence), but it’s very brief and does not fully engage with the literature on KT. For instance, do they mean a form of basic KT [, ,] or integrated KT [, ,] or something else? 5. The description in the introductory section of what is known in the research literature is quite brief. 6. The introductory section also lacks a good description of the gap in the research or what is not known. Also missing is the research question and the intended contribution. What is the paper’s purpose in addressing a gap in our knowledge of KT? 7. A small point, but it would be good to include a description of the paper’s structure in the introduction. 8. In line 43 on page 3 there is reference to “policymaking spaces like technical working groups”. There is a small body of literature that explores the role of space and place in knowledge translation and it should be referenced in the paper, perhaps to identify gaps in this literature and how this study addresses those gaps. 9. As to the research design, why was 17 interviews deemed to be sufficient and 11 to be the saturation point for policy maker interviews? 10. On page 6 in line 46 there is reference to “institutional support”. What specifically is meant by this phrase? 11. The authors do a nice job of organizing the paper around the conceptual framework introduced on page 4. The results on pages 6 to 11 are presented in the context of the framework. But what is missing is a rationale for why this framework was used. What is the gap in our knowledge that is being addressed by using this research framework in this study? 12. I appreciated the quotes provided in the results section, but would have preferred fewer quotes or more narrative presentation of the results of the thematic analysis. 13. The contributions presented in the discussion section do not appear novel. The authors note that previous studies address the benefits of briefs. There is already a literature on the importance of relationships in KT. Perhaps there is a novel contribution in this study regarding the institutionalisation and embedding of KT (see point 1 above as well). However, these topics and other concepts identified in the abstract are not well explored in the discussion section. There may be a novel contribution if the authors examine and develop these concepts in light of the empirical results. 14. On page 7 at line 25 there is reference to more training and workshops. What specifically is meant by these terms? What training and what workshops? 15. The paper will benefit from copy editing. I noted the following items. 16. The paper switches between present and past tenses. 17. Sometimes Oxford commas are used and sometimes not. For instance, on page 3 the Oxford comma is used in the following sentence: “KT broadly involves the synthesis, dissemination, and
--	---

	application of evidence.” The Oxford comma is not used on this sentence on page 4: “This task force draws its membership from the MoH and other government agencies, research and academic institutions, United Nations agencies, civil society organisations and the private sector.” 18. Page 4: The phrase “Task Force” is capitalized in some sentences and not in others. 19. In line 19 on page 4, should the word “advise” be “advises”? 20. Capitalization is applied inconsistently in the section titles. 21. In lists that begin with semi-colons, commas are used in some parts of sentences and in others semi-colons are used, e.g. line 4 on page 8. 22. Thank you again for the opportunity to review and provide comments on this important area of research and practice. I encourage the authors to further develop the concepts in this paper so as to advance our understanding of KT in health care.
--	--

REVIEWER	Margaret Watson University of Strathclyde, Strathclyde Institute of Pharmacy and Biomedical Sciences
REVIEW RETURNED	05-Feb-2022

GENERAL COMMENTS	This study reports a series of interviews with researchers and policymakers and their perceptions of knowledge translation activities during a defined period of the COVID-19 pandemic. Whilst the results might be of interest/relevance at a very local level, due to the small number of participants, particularly the researchers (n=6) as well as their obvious potential bias in the study i.e. researchers as participants, the conclusions that can be derived are relatively limited. It would have been useful if the authors could have included the Framework to which they referred so that the results of could be 'mapped' to the different components. The results section makes references to evidence briefs, however, the relevance of these to the interviews and individual interviewees is not stated. The authors should reflect upon whether their results are robust due to the dual nature of the researchers as study participants. What is the overall learning that can be derived from these interviews and to what extent are they of relevance to a wider audience i.e. beyond the researchers and policy makers who were interviewed? It would be helpful if the authors would reflect upon this final point in particular.
--

VERSION 1 – AUTHOR RESPONSE

Reviewer 1	Response
1. I suggest that the authors discuss the recommendation in more detail - is there anything that would need to change in a KT platform as practiced now or when institutionalized that would enable faster feeding into the emergency response?	Thank you. KT platforms in African countries are significantly lacking or non-existent. Unfortunately, it is not a matter of changing or improving the platforms but rather advocating for their formation, their importance and its institutionalisation. Our study can be used a case study and offers the proof of principle for the establishment of institutionalised KT to respond to emergencies. This is discussed further in the discussion section.

Reviewer 2	Response
1. In the abstract I wanted to know what was meant by “institutionalise KT”, “fully embed KT”, “core functions of research” and “structures that support KT”. I appreciate that these terms may not be able to be defined in the abstract given word limits, but I wanted to see them defined and examined in the paper.	Thank you. We have reviewed this as in this study’s context, all these terms imply institutionalisation of KT.
2. In the introductory section I expected to learn about the relevance of the research to theory and practice. The stated relevance of the study is to address issues with the rapid increase in volume of COVID-19 evidence and misinformation. I did not think a strong case was made for these issues.	Thank you. The relevance of the study is to contribute to the limited literature on the use of evidence during decision making in public health emergencies using KT in Africa. This has now been addressed in the introduction section.
3. I also expected to see a definition of the core construct of knowledge translation (KT). One is given (synthesis, dissemination, and application of evidence), but it’s very brief and does not fully engage with the literature on KT. For instance, do they mean a form of basic KT [1,2,3] or integrated KT [4,5,6] or something else?	Thank you. While different definitions of KT are used, in this study we used the WHO definition which defines KT as “the synthesis, exchange, and application of knowledge by relevant stakeholders to accelerate the benefits of global and local innovation in strengthening health systems and improving people’s health”. In this study we refer to KT exchange models that emphasizes bi-directional interactions between researchers and policy makers. This included both push and exchange models which offer more effective and sustained enhancement of evidence informed policy making. This has been added to the introduction section.
4. The description in the introductory section of what is known in the research literature is quite brief.	Thank you. This has been revised to match the scope of the introduction.
5. The introductory section also lacks a good description of the gap in the research or what is not known. Also missing is the research question and the intended contribution. What is the paper’s purpose in addressing a gap in our knowledge of KT?	Thank you. The gap this study is trying to address is the limited evidence on the use of KT to support decision making during public health emergencies. This has now been added to the introduction section. “While there have been a number of studies on KT in Africa, the literature on the use of KT in responding to PHEs in Africa is limited. More research is needed to inform how institutions in Africa can conduct KT effectively in emergency settings.” And “The lessons learned from these experiences can be integrated into strengthening the processes that promote use of evidence that

	can inform responses during PHEs in African settings.”
6. A small point, but it would be good to include a description of the paper’s structure in the introduction.	The paper has followed structure prescribed by the journal. Stating this in the introduction may not be entirely necessary in our view
7. In line 43 on page 3 there is reference to “policymaking spaces like technical working groups”. There is a small body of literature that explores the role of space and place in knowledge translation and it should be referenced in the paper, perhaps to identify gaps in this literature and how this study addresses those gaps.	Thank you. This study aimed to describe the experiences of researchers and policy makers involved in KT during COVID-19. The KT intervention described and evaluated in this study is the synthesis and communication of evidence through knowledge products such as evidence briefs and summaries. We did not seek to study or evaluate the role of policy making spaces such as technical working groups. This was just one of the interventions implemented separately through EEVIDEM along with reciprocal secondment. These will be evaluated and presented in a different publication.
8. As to the research design, why was 17 interviews deemed to be sufficient and 11 to be the saturation point for policy maker interviews?	Thank you. The study selected all (6) researchers actively engaged in KT in the research institution. While the study identified 11 eligible policy makers to include in the study, conceptual saturation was arrived after interviewing 11 policy makers. That is, by the 11 th interview, there was no new information coming from the interviews. This makes a total of 17 interviews. Our participant groups were relatively homogeneous, and our research aims were thematically focused which meant that information saturation was reached with fewer samples.
9. On page 6 in line 46 there is reference to “institutional support”. What specifically is meant by this phrase?	Thank you. This refers to the support offered by the KEMRI-Wellcome Trust research programme. This has been updated to: “support of and existing KT capacity in KWTRP through EEVIDEM”
10. The authors do a nice job of organizing the paper around the conceptual framework introduced on page 4. The results on pages 6 to 11 are presented in the context of the framework. But what is missing is a rationale for why this framework was used. What is the gap in our knowledge that is being addressed by using this research framework in this study?	Thank you. The framework used here was developed based on the main facilitators of KT derived from literature and underpin KWTRP’s policy engagement and KT strategy. The framework was not intended to address a gap but rather to evaluate the outcomes of our KT activities.
11. I appreciated the quotes provided in the	Thank you. Some of the quotes that were

results section, but would have preferred fewer quotes or more narrative presentation of the results of the thematic analysis.	deemed superfluous upon review have been removed
12. The contributions presented in the discussion section do not appear novel. The authors note that previous studies address the benefits of briefs. There is already a literature on the importance of relationships in KT. Perhaps there is a novel contribution in this study regarding the institutionalisation and embedding of KT (see point 1 above as well). However, these topics and other concepts identified in the abstract are not well explored in the discussion sections. There may be a novel contribution if the authors examine and develop these concepts in light of the empirical results.	Thank you. The importance of institutionalisation in light of our study has been discussed further in the discussion section.
13. On page 7 at line 25 there is reference to more training and workshops. What specifically is meant by these terms? What training and what workshops?	Thank you. The trainings and workshops referred to here are on KT and evidence informed decision making. This has been clarified in the text.
14. The paper switches between present and past tenses.	Thank you. This has been addressed.
15. Sometimes Oxford commas are used and sometimes not. For instance, on page 3 the Oxford comma is used in the following sentence: "KT broadly involves the synthesis, dissemination, and application of evidence." The Oxford comma is not used on this sentence on page 4: "This task force draws its membership from the MoH and other government agencies, research and academic institutions, United Nations agencies, civil society organisations and the private sector."	Thank you. This has been revised.
16. Page 4: The phrase "Task Force" is capitalized in some sentences and not in others.	Thank you. This has been revised.
17. In line 19 on page 4, should the word "advise" be "advises"?	Thank you. This has been corrected
18. Capitalization is applied inconsistently in the section titles.	Thank you. This has been revised.

19. In lists that begin with semi-colons, commas are used in some parts of sentences and in others semi-colons are used, e.g. line 4 on page 8.	Thank you. This has been revised.
Reviewer 3	Response
1. Whilst the results might be of interest/relevance at a very local level, due to the small number of participants, particularly the researchers (n=6) as well as their obvious potential bias in the study i.e. researchers as participants, the conclusions that can be derived are relatively limited.	Thank you. The reviewer raises two valid concerns. One, on the generalizability of the findings and the dual role of researchers as participants. On generalizability: We have not highlighted that this qualitative study, consistent with other qualitative studies did not aim at statistical generalizability (which cannot be achieved by a small non-random sample). Rather we aimed at analytical generalizability: to explore and elucidate themes that might relevant and considered for exploration in the study setting and in other setting similar to the study setting. Analytical generalizability, unlike statistical generalizability can be achieved by small, purposive samples that typically characterize qualitative studies (1). This analysis can be considered as a useful case study of the successful implementation of KT interventions during a public health emergency. Even though many of the issues explored are of relevance to evidence informed decision-making more generally, it may also provide insights into the utility of KT processes to respond to crises. The study highlights how an examination of the engagement between researchers and policy makers and the uptake of evidence in a pandemic can contribute to evidence informed policy making. On the dual role of researchers as participants: this is addressed in comment 4 below
2. It would have been useful if the authors could have included the Framework to which they referred so that the results of could be 'mapped' to the different components.	Thank you. The adopted framework has been included (page 4) and the results have been mapped/discussed in the context of the framework from pages 6-11
3. The results section makes references to evidence briefs, however, the relevance of these to the interviews and individual interviewees is not stated.	Thank you. The evidence briefs were the communication tool used in our KT process. Consequently, participants were asked to comment on their usefulness. The interviewees agreed that they were an efficient tool for delivery of evidence and its relevance was in how it made evidence accessible to policy makers. This is

	highlighted in the results section under accessibility (pg. 5)
4. The authors should reflect upon whether their results are robust due to the dual nature of the researchers as study participants	Thank you. We acknowledge that researchers as participants may have impacted our analysis and findings. However, the participatory nature of our KT research and processes makes it inherently collaborative which inadvertently sets the researcher up as a participant as well. To ensure validity, the researchers were transparent about their positioning, their relationship to the topic of the investigation and maintained reflexivity throughout the process. In addition, the analysis was presented to the participants for sense checking to see if their views were correctly represented. The study was also presented to wider forums to check for objectivity and critique. Finally, some of the findings reported are consistent with findings in other studies. We believe that these factors speak to the robustness of our findings. This has been added to the limitation section of the manuscript.
5. What is the overall learning that can be derived from these interviews and to what extent are they of relevance to a wider audience i.e. beyond the researchers and policy makers who were interviewed? It would be helpful if the authors would reflect upon this final point in particular.	Thank you. The key learning from this study is the importance of institutionalising KT in both research/academic and government institutions to support public health emergencies. In this study, we sought to increase evidence uptake by focusing on enhancing 4 main determinants of uptake: accessibility, motivation, capacity and relationships. We found that by using this framework, we had relative success in influencing policy making during COVID-19. Though this approach may not be novel, the context within which it is applied is. While there have been a number of studies on the challenges of conducting KT in Africa, very little has been done on how institutions in Africa can respond to complex public health emergencies. It was therefore important to extract the lessons we learned from COVID-19 by accurately recording our experiences, interventions and outcomes to inform the establishment of institutionalised KT. This has been included in the conclusion.

VERSION 2 – REVIEW

REVIEWER	Tina Purnat World Health Organization, Department of Pandemic and Epidemic Preparedness and Prevention
REVIEW RETURNED	21-Mar-2022

GENERAL COMMENTS	The authors addressed the comments from the previous review.
--

REVIEWER	David Foord University of New Brunswick
-----------------	--

REVIEW RETURNED	23-Mar-2022
-------------

GENERAL COMMENTS	Thank you again for the opportunity to comment. I thank the researchers for this revised version of the manuscript. The main question I have is what is meant by institutionalization? This concept is central to the manuscript's results and conclusion sections, as well as the author's contribution to theory and practice. On page 3 of the manuscript, the authors state that "Institutionalising KT means taking steps to ensure that the systems, structures, resources and support needed for KT to become standard practices are available." The resources and support are well described in the paragraph, but the paragraph is not clear on the systems and structures. May they be described in the paper? How specifically did they develop and implement KT strategies and coordinate KT in the institution? Perhaps this can be addressed on pages 11 and 12. Institutionalization is also discussed on pages 11 and 12. In addition to question above about systems and structures, two other questions came to mind in reading this section. 1. The authors state there is an urgent need to institutionalise KT within research and government institutions in Africa to respond to PHEs adequately? How does or how might the institutionalization of KT differ in research and government institutions? Or is there no difference in the systems, structures, resources and support in these two locations? 2. The following two sentences on page 11 raise interesting questions about the relationship between research and KT. It also provides an opportunity to make a novel contribution to global theory and practice of KT. "Institutions can hire more KT experts to provide on-demand evidence promptly. This would put less pressure on researchers who balance their daily research activities while responding to new evidence demands." The focus of much of the KT literature and this paper is on the relationships between researchers and policy makers / practice. Can the authors tell us more about the relationship between researchers and KT experts, as well as the specific division of labour between the two? The manuscript will still benefit from copy editing, e.g. in the title, should the word "Knowledge" be lowercase? I note that the text in the paper still switches between present and past tenses.
---

VERSION 2 – AUTHOR RESPONSE

Reviewer 2	Response
1. The main question I have is what is meant by institutionalization? This concept is central to the manuscript's results and conclusion sections, as well as the author's contribution to theory and practice. On page 3 of the manuscript, the authors state that "Institutionalising	Thank you. Several questions have been raised: - How specifically did they develop and implement KT strategies and coordinate KT in the institution? KEMRI-Wellcome programme developed and implemented the KT strategies

KT means taking steps to ensure that the systems, structures, resources and support needed for KT to become standard practices are available.” The resources and support are well described in the paragraph, but the paragraph is not clear on the systems and structures. May they be described in the paper? How specifically did they develop and implement KT strategies and coordinate KT in the institution? Perhaps this can be addressed on pages 11 and 12.	through the EEVIDEM project described in pages 2-3. The objective of this project was to institutionalise KT within the research programme by providing the resources needed to carry out KT. The programme’s KT efforts were implemented and evaluated by the KT team that was formed under EEVIDEM. This included collating and synthesizing evidence, developing knowledge products and dissemination of these products (pg 3-4). EEVIDEM also supported 3 other interventions including facilitating the embedding of researchers in policy spaces. These other interventions are discussed in a separate manuscript. - The resources and support are well described in the paragraph, but the paragraph is not clear on the systems and structures. May they be described in the paper? The systems and structures mentioned here include the development of a policy engagement strategy document for the programme, mechanisms for routine documentation and evaluation of KT and developing structures that facilitate linkages between researchers and policy makers such as the formation of an evidence synthesis group which is made up of researchers and policy makers to co-produce evidence. This has been added to the paragraph on pg 3.
2. The authors state there is an urgent need to institutionalise KT within research and government institutions in Africa to respond to PHEs adequately? How does or how might the institutionalization of KT differ in research and government institutions? Or is there no difference in the systems, structures, resources and support in these two locations?	Thank you. The institutionalisation of KT within research and policy institutions is similar in terms of resources and structures required. Both institutions require significant investments to support KT by building capacity and sustaining KT efforts. Structures that facilitate linkages between the two types of institutions are also required (e.g establishing memorandums of understandings).
3. The following two sentences on page 11 raise interesting questions about the relationship between research and KT. It also provides an opportunity to make a novel contribution to global theory and practice of KT. “Institutions can hire more KT experts to provide on-demand evidence promptly. This would put less pressure on researchers who balance their daily research activities while responding to new evidence demands.” The focus of much of the KT literature and this paper is on the relationships	Thank you. The role that KT experts play and the relationships between KT experts and researchers is an important area to explore. However, in this study that relationship was not explored as we focused more on the relationship between policy makers and researchers. We therefore feel ill-equipped to comment on the dynamics of that relationship in our context.

between researchers and policy makers / practice. Can the authors tell us more about the relationship between researchers and KT experts, as well as the specific division of labour between the two?	
4. The manuscript will still benefit from copy editing, e.g. in the title, should the word "Knowledge" be lowercase? I note that the text in the paper still switches between present and past tenses.	Thank you. This has been revised.

VERSION 3 – REVIEW

REVIEWER	David Foord University of New Brunswick
REVIEW RETURNED	19-Apr-2022
GENERAL COMMENTS	The reviewer completed the checklist but made no further comments.